# Characterizing Optimal Mixed Policies:
# Where to Intervene and What to Observe

**Sanghack Lee**    **Elias Bareinboim**
Causal Artificial Intelligence Laboratory
Columbia University
{sanghacklee,eb}@cs.columbia.edu

## Abstract

Intelligent agents are continuously faced with the challenge of optimizing a policy based on what they can observe (see) and which actions they can take (do) in the environment where they are deployed. Most policies can be parametrized in terms of these two dimensions, i.e., as a function of what can be seen and done given a certain situation, which we call a *mixed policy*. In this paper, we investigate several properties of the class of mixed policies and provide an efficient and effective characterization, including optimality and non-redundancy. Specifically, we introduce a graphical criterion to identify unnecessary contexts for a set of actions, leading to a natural characterization of non-redundancy of mixed policies. We then derive sufficient conditions under which one strategy can dominate the other with respect to their maximum achievable expected rewards (optimality). This characterization leads to a fundamental understanding of the space of mixed policies and a possible refinement of the agent's strategy so that it converges to the optimum faster and more robustly. One surprising result of the causal characterization is that the agent following a more standard approach—intervening on all intervenable variables and observing all available contexts—may be hurting itself, and will never achieve an optimal performance.

## 1 Introduction

Agents are deployed in complex and uncertain environments where they are bombarded with high volumes of information and are expected to operate efficiently, safely, and rationally. The discipline of causal inference (CI) offers a compelling set of tools and a language that allows one to reason with the structural invariances present in complex environments [1–5]. Whenever the causal mechanisms of an underlying environment are sufficiently well-understood, the agent can design very precise interventions, bringing a certain desired state of affairs about swiftly and cleanly (e.g., personalized medical treatments, inequality-reducing tax policies). In the field of ML, bandits and reinforcement learning (RL) constitute the *de facto* framework in which agents are designed such that a certain policy is optimized and the corresponding goals can be efficiently achieved [6–8].

There is a growing literature exploring how these two frameworks (RL and CI) are related, and how this understanding can be translated into more efficient decision-making in more challenging and realistic settings. Recently, the more explicit connection between these frameworks has been made by eliciting how causal knowledge—unobserved confounders and the causal relations between actions, contexts, and rewards—can be used to improve decision-making in a variety of settings, including for both interventional [9–11] and counterfactual [12, 13] reasoning (see also [14–17] and [18–21]). Outside more traditional RL, causal inference researchers have embraced the idea of sequential decision making in terms of conditional plans or dynamic treatment regimes, while focusing on, e.g., the identifiability of causal effects from observational data [22–27].

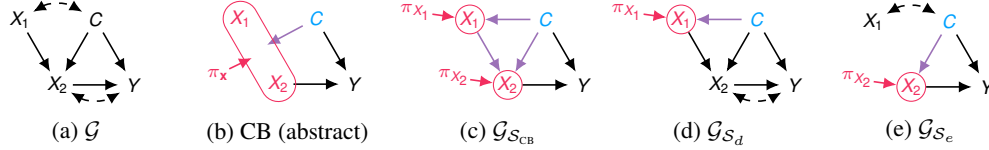

(a) $\mathcal{G}$  (b) CB (abstract)  (c) $\mathcal{G}_{\mathcal{S}_{\text{CB}}}$  (d) $\mathcal{G}_{\mathcal{S}_d}$  (e) $\mathcal{G}_{\mathcal{S}_e}$

Figure 1: (a) a causal graph, (b) abstract representation of a contextual bandit policy, and (c,d,e) policy-induced graphs. Red circles for the intervened variables and, as a supplement, blue for their non-action contexts and purple for induced edges (i.e., contexts *cause* an action). $\pi$ nodes are intervention indicators, which will be left implicit throughout the paper.

One of the main tasks in decision-making is to optimize the parameters associated with a specific policy. The scope of each policy is usually fixed, in the sense that the set of actions and contexts are pre-specified, *a priori*. By and large, the literature considers policies with scope that is (1) observational, where the system is allowed to evolve without any intervention; (2) fully experimental, where all the *action* variables are intervened on and all the *context* variables are observed. The former tends to be more common in CI while the latter tends to be more common in RL. A causal understanding of the world gives rise to a rich spectrum of policies with different scopes, allowing agents to choose how to interact with the environment, meaning, which variables to intervene and to observe (as a context). Against this background, we consider exploiting causal relationships for systematic decision making in the context of, so called, *mixed policies*, which consists of a set of decision rules where each rule corresponds to the way an action for an intervenable variable is determined given its contexts.

For concreteness, consider an agent deployed in an environment represented as a *causal graph* $\mathcal{G}$ (Fig. 1a), where $C, \mathbf{X} = \{X_1, X_2\}, Y$ represent the context, two action variables, and the reward variable, respectively. Graphically, bidirected edges roughly represent unobserved confounders (UCs, for short) affecting both ends of the arrow. The agent's task is to maximize the reward $\mu_{\boldsymbol{\pi}} \doteq \mathbb{E}_{\boldsymbol{\pi}}[Y]$ under a mixed policy (or simply, policy) $\boldsymbol{\pi} \in \boldsymbol{\Pi}$, where $\boldsymbol{\Pi}$ is a mixed policy space. A mixed policy is associated with its *scope*, called *mixed policy scope* (MPS), which specifies the variables the policy are intervening, and the variables taken into account for each intervened variables.

A standard contextual bandit (CB) optimizes a policy $\boldsymbol{\pi}_{\text{CB}}$ (Fig. 1b), a (stochastic) mapping from contexts to actions, which can be equally represented as a pair of decision rules $\boldsymbol{\pi}_{\text{CB}} = \{\pi(x_1|c), \pi(x_2|x_1, c)\}$ (Fig. 1c). Traditionally, the policy is optimized within a restricted space $\boldsymbol{\Pi}_{\text{CB}}$, characterized by policies following a scope $\mathcal{S}_{\text{CB}} = \{\langle X_1, \{C\}\rangle, \langle X_2, \{X_1, C\}\rangle\}$ that $X_1$ is determined by $C$ and $X_2$ is decided based on $C$ and $X_1$. Unfortunately, the optimal policy $\boldsymbol{\pi}_{\text{CB}}^* \doteq \arg\max_{\boldsymbol{\pi} \in \boldsymbol{\Pi}_{\text{CB}}} \mu_{\boldsymbol{\pi}}$ can be suboptimal, i.e., $\mu_{\mathcal{S}_{\text{CB}}}^* \doteq \mu_{\boldsymbol{\pi}_{\text{CB}}^*} < \mu^*$ where $\mu^*$ is the optimal expected reward. To ground what this means, let every variable be binary and $U_1$ and $U_2$, the UCs adjacent to $X_1$ and $X_2$, be fair coins and $\epsilon$ be a noise over $X_1$ following $P(\epsilon = 1) = 0.2$. Also, let the unobserved causal mechanisms be specified as $X_1 \leftarrow U_1 \oplus \epsilon$, $C \leftarrow U_1$, $X_2 \leftarrow U_2 \oplus X_1 \oplus C$, and $Y \leftarrow (1 - (X_2 \oplus U_2)) \vee C$, where $\oplus$ is the exclusive-or operator. Since the policy determines $X_2$ irrelevant to $U_2$ and the context $C$ is also independent to $U_2$, we can elicit that $\mu_{\mathcal{S}_{\text{CB}}}^* = 0.75$. In this setting, the best policy is intervening only on $X_1$ given $C$, i.e., $\{\pi(x_1|c)\}$ as depicted in Fig. 1d. With $X_1 = C$, the policy suppresses the noise $\epsilon$ over $X_1$ and makes $X_2 = U_2$ so that its optimal expected reward in this environment is 1.0.

In the example of Fig. 1a, if $\{X_1, X_2\}$ are intervenable and $\{C, X_1\}$ can become a context, there are 15 mixed policy scopes. These different modes of interaction can be represented as induced graphs and can be classified based on two desiderata: *non-redundancy* and *optimality*. We explain these desiderata through an illustration (Fig. 2) of the four MPSes $\mathcal{S}_a = \{\}, \mathcal{S}_{\text{CB}}, \mathcal{S}_d = \{\langle X_1, \{C\}\rangle\}$, and $\mathcal{S}_e = \{\langle X_2, \{C\}\rangle\}$. We annotate their relationships with a superset symbol $\supset$, whether one scope has more actions or contexts than the other, and with a comparison symbol $\geq_\mu$ (or $=_\mu$), whether one's optimal reward is at least as good as (or equal to) the other's in *every* world compatible with a causal graph. The equality forms an equivalence class among scopes with respect to optimal rewards.

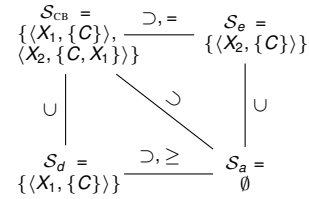

Figure 2: Relationships between the MPSes

Roughly speaking (to be formalized later on), *non-redundancy* refers to the condition of a scope

such that removing any of its actions or contexts can negatively affect its maximum performance. In other words, given two scopes $\mathcal{S}$ and $\mathcal{S}'$, if $\mathcal{S} \supsetneq \mathcal{S}'$ and $\mathcal{S} =_\mu \mathcal{S}'$, then $\mathcal{S}$ is said to be redundant. For instance, since $\mathcal{S}_c \supset \mathcal{S}_e$ while $\mathcal{S}_c =_\mu \mathcal{S}_e$, the CB policy (Fig. 1c) is redundant and the CB agent wastes its resources not only for intervening on $X_1$ (a redundant action) but also for taking $X_1$ into account for $X_2$ (a redundant context). Furthermore, *optimality* of a scope $\mathcal{S}$ represents that there exists no other scope $\mathcal{S}'$ (not in the equivalence class of $\mathcal{S}$) such that $\mathcal{S}' \geq_\mu \mathcal{S}$. For example, $\mathcal{S}_d$, when optimized, is at least as good as $\mathcal{S}_a$ (i.e., $\mu_{\mathcal{S}_d}^* \geq \mu_{\mathcal{S}_a}^*$) in every environment, and can outperform it in some environments (i.e., $\mu_{\mathcal{S}_d}^* > \mu_{\mathcal{S}_a}^*$), which demonstrates that $\mathcal{S}_a$ does not meet the optimality criterion. Not every pair of scopes can be comparable: $\mathcal{S}_e$ is not comparable to $\mathcal{S}_a$ nor $\mathcal{S}_d$. After a careful examination, we can indeed able to show that MPSes $\mathcal{S}_c, \mathcal{S}_d, \mathcal{S}_e$ meet the optimality criterion. Both non-redundancy and optimality are satisfied only by $\mathcal{S}_d$ and $\mathcal{S}_e$ among all 15 scopes. This example demonstrates that an intelligent agent should judiciously intervene on a carefully chosen subset of variables with side information (context) relevant to attaining an optimal reward. More detailed account is given in Appendix A [28].

**Contributions**  In this work, we investigate mixed policies with respect to their expected rewards. Our contributions are as follows. (i) We developed a graphical criterion that detects the redundancy of contexts relative to a collection of actions taking advantage of properties pertain to optimal mixed policies. (ii) We established sufficient conditions under which one policy scope can outperform another, characterizing the partial order defined over the space of scopes with respect to their maximum expected rewards achievable. We believe these results have practical implications for the design of intelligent agents providing the basis for efficient and effective explorations of the policy space. One fundamental implication of our analysis is that the agent following a standard approach (i.e., intervening and observing whenever possible) may be hurting itself, and, regardless of the number of interactions, will never be able to achieve an optimal performance.

**Preliminaries**  Let us denote a variable by an uppercase letter $X$, whose value is denoted by its corresponding lowercase letter $x$. A set of variables will be denoted by a bold uppercase letter $\mathbf{X}$ with its value $\mathbf{x}$. We follow notational conventions from literature on measure theory, algebra of sets, and causal inference. We may use $\dot{\cup}$, instead of $\cup$, to emphasize the union of two disjoint sets. We use structural causal models (SCMs) [1, Ch. 7] as the semantic framework to represent an underlying environment. An SCM $\mathcal{M}$ is a quadruple $\langle \mathbf{U}, \mathbf{V}, P(\mathbf{U}), \mathbf{F} \rangle$, where $\mathbf{U}$ is a set of exogenous variables determined by factors outside the model following a joint distribution $P(\mathbf{U})$, and $\mathbf{V}$ is a set of endogenous variables whose values are determined following a collection of functions $\mathbf{F} \doteq \{f_i\}_{V_i \in \mathbf{V}}$ such that $V_i \leftarrow f_i(\mathbf{pa}_i, \mathbf{u}_i)$ where $\mathbf{PA}_i \subseteq \mathbf{V} \backslash \{V_i\}$ and $\mathbf{U}_i \subseteq \mathbf{U}$. The observational distribution $P(\mathbf{v})$ is defined as $\sum_{\mathbf{u}} \prod_{V_i \in \mathbf{V}} P(v_i | \mathbf{pa}_i, \mathbf{u}_i) P(\mathbf{u})$. Further, $do(\mathbf{X} = \mathbf{x})$ represents the operation of fixing a set $\mathbf{X}$ to a constant $\mathbf{x}$ regardless of their original mechanisms. Such intervention induces a submodel $\mathcal{M}_\mathbf{x}$, which is $\mathcal{M}$ with $f_X$ replaced to $x$ for $X \in \mathbf{X}$. Then, an interventional distribution $P_\mathbf{x}(\mathbf{v} \backslash \mathbf{x})$ (or also $P(\mathbf{v} \backslash \mathbf{x} | do(\mathbf{x}))$) follows from $\mathcal{M}_\mathbf{x}$, and is such that $P_\mathbf{x}(\mathbf{v} \backslash \mathbf{x}) = \sum_{\mathbf{u}} \prod_{V_i \in \mathbf{V} \backslash \mathbf{X}} P(v_i | \mathbf{pa}_i, \mathbf{u}_i) P(\mathbf{u})$.

Graphically, each SCM (model, for short) is associated with a causal diagram $\mathcal{G} = \langle \mathbf{V}, \mathbf{E} \rangle$, where each type of edge represents a different relationship among variables: (i) $X \rightarrow Y$ if $X$ is an argument of $f_Y$ (a direct causal relationship); and (ii) $X \leftrightarrow Y$ if for a maximal subset $\mathbf{W} \subseteq \mathbf{V} \backslash \{X\}$ such that $\mathbf{U}_\mathbf{W} \perp\!\!\!\perp \mathbf{U}_X$ and $\mathbf{U}_Y \not\subseteq \mathbf{U}_\mathbf{W}$; From the agent's perspective, only the causal graph $\mathcal{G}$ of the environment $\mathcal{M}$ is available, while its reward is validated through $\mathcal{M}$. We operate in the non-parametric setting, where no assumption about the form or shape of the pair $\langle P(\mathbf{U}), \mathbf{F} \rangle$ is made, but for the structural knowledge encoded in $\mathcal{G}$. Whenever not even $\mathcal{G}$ is known, the agent can perform active interventions to learn it; for example, see [29, 30]. We denote by $\mathcal{G}_{\overline{\mathbf{X}}\underline{\mathbf{Z}}}$ an edge subgraph of $\mathcal{G}$ which removes edges incoming to $\mathbf{X}$ and outgoing from $\mathbf{Z}$. A submodel $\mathcal{M}_\mathbf{x}$ can be presented as $\mathcal{G}_{\overline{\mathbf{X}}}$ with $\mathbf{X}$ fixed to $\mathbf{x}$. Hence, causal relationships among other variables are captured in $\mathcal{G} \backslash \mathbf{X}$, which is the subgraph of $\mathcal{G}$ over $\mathbf{V} \backslash \mathbf{X}$. We denote by $\mathcal{G} \langle \mathbf{V}' \rangle$ the *latent projection* of $\mathcal{G}$ onto $\mathbf{V}'$, the causal graph retaining causal relationships among $\mathbf{V}'$ [31]. We adopt familial notation, $ch, pa, an, de$ for children, parents, ancestors, and descendants, respectively, with $Ch, Pa, An, De$ including arguments. Our work utilizes d-separation [32, 33] and do-calculus [34], classic graphical rules to ascertain equalities between distributions. The omitted proofs and derivations are provided in [28].

## 2  Mixed Policies: Fundamentals & Basic Results

As discussed in the previous section, a causal understanding of the underlying world helps recognize a broad spectrum of policies with diverse scopes so as for agents to select the mode of interaction. We now formally define the space of mixed policies with the notion of mixed policy scope.

**Definition 1** (Mixed Policy Scope (MPS)). *Let $\mathcal{G}$ be a causal graph, $Y$ be a specific reward variable, $\mathbf{X}^{\star} \subseteq \mathbf{V} \backslash \{Y\}$ a set of intervenable variables, and $\mathbf{C}^{\star} \subseteq \mathbf{V} \backslash \{Y\}$ a set of contextualizable variables. A mixed policy scope $\mathcal{S}$ is defined as a collection of pairs $\langle X, \mathbf{C}_X \rangle$ such that (i) $X \in \mathbf{X}^{\star}$, $\mathbf{C}_X \subseteq \mathbf{C}^{\star} \backslash \{X\}$, and (ii) $\mathcal{G}_{\mathcal{S}}$ is acyclic, where $\mathcal{G}_{\mathcal{S}}$ is defined as $\mathcal{G}$ with edges onto $X$ removed and directed edges from $\mathbf{C}_X$ to $X$ added for every $\langle X, \mathbf{C}_X \rangle \in \mathcal{S}$.*

For concreteness, given a causal graph $\mathcal{G}$ (Fig. 1a), the observational case is an MPS $\{\}$. An MPS $\mathcal{S}_{\mathrm{CB}} = \{\langle X_1, \{C\} \rangle, \langle X_2, \{X_1, C\} \rangle\}$ induces a graph (Fig. 1c) while $\{\langle X_2, \{C\} \rangle\}$ induces a graph in Fig. 1e. An MPS represents a class of mixed policies that share the same graphical characteristics manifested by $\mathcal{G}_{\mathcal{S}}$, an induced graph for $\mathcal{M}_{\boldsymbol{\pi}}$.

**Definition 2** (Mixed Policy). *Given $\langle \mathcal{G}, Y, \mathbf{X}^{\star}, \mathbf{C}^{\star} \rangle$ and an SCM $\mathcal{M} \sim \mathcal{G}$ with $\mathfrak{X}_Y \subseteq \mathbb{R}$, a mixed policy $\boldsymbol{\pi}$ is a realization of a mixed policy scope $\mathcal{S}$ compatible with the tuple $\boldsymbol{\pi} \doteq \{\pi_{X|\mathbf{C}_X}\}_{\langle X, \mathbf{C}_X \rangle \in \mathcal{S}}$, where $\pi_{X|\mathbf{C}_X} : \mathfrak{X}_X \times \mathfrak{X}_{\mathbf{C}_X} \mapsto [0, 1]$ is a proper probability mapping.*

If we consider the MPS $\mathcal{S}_{\mathrm{CB}}$ discussed above, its mixed policy $\boldsymbol{\pi}$ is $\{\pi_{X_1|\{C\}}, \pi_{X_2|\{C,X_1\}}\}$, which is a specific instantiation of the parameters with respect to the corresponding scope. For readability, we may write $\{\pi(x_1|c), \pi(x_2|x_1, c)\}$. Given an underlying SCM $\mathcal{M}$, a mixed policy $\boldsymbol{\pi}$ induces a variant of SCM $\mathcal{M}_{\boldsymbol{\pi}}$ where the function for $X \in \mathbf{X}(\boldsymbol{\pi})$ is replaced by the corresponding $\pi_{X|\mathbf{C}_X}$ (see [35] for a detailed account). We denote by $P_{\boldsymbol{\pi}}$ the joint distribution over the variables from the system under the policy $\boldsymbol{\pi}$. Throughout the paper, $\mathcal{G}$, $Y$, $\mathbf{C}^{\star}$, and $\mathbf{X}^{\star}$ are oftentimes implicit including an underlying SCM $\mathcal{M} \sim \mathcal{G}$ and, thus, $\boldsymbol{\Pi}$, as well.

**Expected Reward**   We define the expected reward of a mixed policy. To begin with, we define intervened variables $\mathbf{X}(\mathcal{S}) \doteq \{X \mid \langle X, \mathbf{C}_X \rangle \in \mathcal{S}\}$ and active contexts $\mathbf{C}(\mathcal{S}) \doteq \bigcup_{X \in \mathbf{X}(\mathcal{S})} \mathbf{C}_X$. Similarly, given $\boldsymbol{\pi} \sim \mathcal{S}$ (a mixed policy following the MPS), $\mathbf{X}(\boldsymbol{\pi}) \doteq \mathbf{X}(\mathcal{S})$ and $\mathbf{C}(\boldsymbol{\pi}) \doteq \mathbf{C}(\mathcal{S})$. Let $\mathbf{C}^- = \mathbf{C}(\boldsymbol{\pi}) \backslash \mathbf{X}(\boldsymbol{\pi})$ be the *non-action* contexts. Then, the expected reward for $\boldsymbol{\pi}$ can be expressed as, with $\mathbf{x}$ simply denoting the value of $\mathbf{X}(\mathcal{S})$,

$$\mu_{\boldsymbol{\pi}} = \sum_{y, \mathbf{x}, \mathbf{c}^-} y P_{\mathbf{x}}(y, \mathbf{c}^-) \prod_{X \in \mathbf{X}(\boldsymbol{\pi})} \pi(x|\mathbf{c}_x). \tag{1}$$

The expression separates the atomic interventional probability (first factor), which is inherent to the underlying world and not affected by the policy $\boldsymbol{\pi}$, from the likelihood of a specific intervention given contexts (second factor), which is optimizable and defined by $\boldsymbol{\pi}$. The expected reward can also be written focusing only on a subset of intervened variables. Given $\mathbf{X}' \subseteq \mathbf{X}(\boldsymbol{\pi})$, let $\mathbf{C}' = \bigcup_{X \in \mathbf{X}'} \mathbf{C}_X \backslash \mathbf{X}'$, and $Q' = P_{\boldsymbol{\pi} \backslash \mathbf{X}'}$ where $\boldsymbol{\pi} \backslash \mathbf{X}'$ represents $\boldsymbol{\pi}$ with decision rules over $\mathbf{X}'$ removed. Then, $\mu_{\boldsymbol{\pi}} = \sum_{y, \mathbf{x}', \mathbf{c}'} y Q'_{\mathbf{x}'}(y, \mathbf{c}') \prod_{X \in \mathbf{X}'} \pi(x|\mathbf{c}_x)$. This expression, which hides the details of uninteresting actions and contexts, is the building block to characterize mixed policies.

**Optimality and deterministic mixed policy**   A mixed policy $\boldsymbol{\pi}$ is said to be *optimal* in the given environment if and only if $\mu_{\boldsymbol{\pi}} = \mu^* \doteq \max_{\boldsymbol{\pi}' \in \boldsymbol{\Pi}} \mu_{\boldsymbol{\pi}'}$. Restricting our attention to $\boldsymbol{\Pi}_{\mathcal{S}} \doteq \{\boldsymbol{\pi} \in \boldsymbol{\Pi} \mid \boldsymbol{\pi} \sim \mathcal{S}\}$, we define $\mu^*_{\mathcal{S}} \doteq \max_{\boldsymbol{\pi}' \in \boldsymbol{\Pi}_{\mathcal{S}}} \mu_{\boldsymbol{\pi}'}$, an optimal policy $\boldsymbol{\pi}$ *with respect to* $\mathcal{S}$. We call a mixed policy *deterministic* if, for every $\pi_{X|\mathbf{C}_X} \in \boldsymbol{\pi}$, $X$ is determined by a function of $\mathbf{C}_X$.

**Proposition 1.** *Given a mixed policy scope, there always exists a deterministic mixed policy, which is optimal with respect to the given scope.*

Not surprisingly at this point, a stochastic policy is no better than the best deterministic policy [36–38]. Still, this result has a particular importance to the treatment provided here due to its implications to the d-separation criterion [39], which will be instrumental and discussed in depth in Sec. 3.1. Another key implication is shown next.

**Proposition 2** (Separation of Actions and Contexts). *Given an MPS $\mathcal{S}$, there always exists a deterministic mixed policy $\boldsymbol{\pi} \in \boldsymbol{\Pi}$ such that $\mathbf{X}(\boldsymbol{\pi})$ and $\mathbf{C}(\boldsymbol{\pi})$ are disjoint and $\mu_{\boldsymbol{\pi}} = \mu^*_{\mathcal{S}}$.*

A deterministic policy gives rise to the autonomy of each action allowing them to be determined only by *non-action* contexts. For concreteness, consider the example shown in Fig. 3a. A mixed policy (Fig. 3b) includes $X_2$ listening to $X_1$, which enables systematic coordination between $X_1$ and $X_2$. The proposition implies that $X_2$ can rather listen to $C$ (which is the context of $X_1$) directly (Fig. 3d). Further, in Fig. 3c, $X_2$ utilizes both $X_1$ and $C$. However, it is sufficient to make use of only $C$. By noting that the policy relative

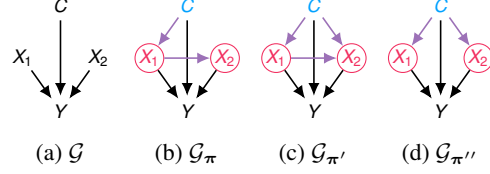

Figure 3: Given a causal graph (a), three induced graphs (b,c,d) for different mixed policies where (d) the separation is demonstrated.

(a) $\mathcal{G}$   (b) $\mathcal{G}_{\boldsymbol{\pi}}$   (c) $\mathcal{G}_{\boldsymbol{\pi}'}$   (d) $\mathcal{G}_{\boldsymbol{\pi}''}$

to Fig. 3d can achieve optimality, while relying on lesser information than the one relative to Fig. 3c, we investigate how to capture non-redundancy within MPSes.

## 3 Non-Redundant Mixed Policy

Optimizing a mixed policy involves assessments of the effectiveness of its scope so that an agent can avoid intervening or observing on unnecessary actions or contexts. Here, we define and characterize non-redundancy of MPS. We say $\mathcal{S}$ subsumes $\mathcal{S}'$, denoted by $\mathcal{S}' \subseteq \mathcal{S}$, if $\mathbf{X}(\mathcal{S}') \subseteq \mathbf{X}(\mathcal{S})$ and $\mathbf{C}'_X \subseteq \mathbf{C}_X$, for every $\langle X, \mathbf{C}'_X \rangle \in \mathcal{S}'$. Further, we denote by $\boldsymbol{\pi}' \subseteq \boldsymbol{\pi}$, where $\boldsymbol{\pi}' \sim \mathcal{S}'$ and $\boldsymbol{\pi} \sim \mathcal{S}$ if $\pi'(x|\mathbf{c}'_x) = \sum_{\mathbf{c}''_x} \pi(x|\mathbf{c}_x) P_{\boldsymbol{\pi}}(\mathbf{c}''_x|\mathbf{c}'_x)$, for every $X \in \mathbf{X}(\mathcal{S}')$ where $\mathbf{C}''_X = \mathbf{C}_X \backslash \mathbf{C}'_X$.

**Definition 3.** Given $\langle \mathcal{G}, Y, \mathbf{X}^\star, \mathbf{C}^\star \rangle$, an MPS $\mathcal{S}$ is said to be *non-redundant* if there exists an SCM $\mathcal{M} \sim \mathcal{G}$ and $\boldsymbol{\pi} \sim (\mathcal{S}, \mathcal{M})$ such that $\mu_{\boldsymbol{\pi}} \neq \mu_{\boldsymbol{\pi}'}$ for every $\boldsymbol{\pi}' \subsetneq \boldsymbol{\pi}$.

The constraint on $\boldsymbol{\pi}'$ ensures that the definition of non-redundancy of MPS is focused on the differences in actions or contexts while the behavior (i.e., $\pi(\cdot|\cdot)$) remains the same—$\pi'(x|\mathbf{c}'_x) = Q(x|\mathbf{c}'_x)$ if $\mathbf{C}'_X \neq \mathbf{C}_X$ and $Q(x|\mathbf{c}_x) = \pi'(x|\mathbf{c}_x) = \pi(x|\mathbf{c}_x)$, otherwise. Hence, the constraint provides a basis to characterize non-redundancy of MPS utilizing well-established graphical criteria.

**Theorem 1.** *Let $\mathcal{S} = \{\langle X, \mathbf{C}_X \rangle\}_{X \in \mathbf{X}}$ be an MPS and let $\mathcal{H} = \mathcal{G}_{\mathcal{S}}$. $\mathcal{S}$ is non-redundant if and only if (i) $\mathbf{X} \subseteq an(Y)_{\mathcal{H}}$ and (ii) $(C \not\perp\!\!\!\perp Y \mid \mathbf{C}_X \backslash \{C\})$ in $\mathcal{H} \backslash \{X\}$, for every $X \in \mathbf{X}$ and $C \in \mathbf{C}_X$.*

The condition (i) can be seen through rule 3 of do-calculus such that the change of the mechanism of $X$ has a consequence on the reward $Y$.[1] The condition (ii) coincides with rule 2 of do-calculus $Q(y|x, \mathbf{c}_x \backslash \{c\}) = Q_x(y|\mathbf{c}_x \backslash \{c\})$, where $Q = P_{\boldsymbol{\pi}}$. In words, the path from $C$ to $Y$ can be concatenated with $X \leftarrow C$ to form a back-door path from $X$ to $Y$.[2] Consider the example in Fig. 4 where both $X_1$ and $X_2$ are ancestors of $Y$ (condition (i)). Regarding condition (ii), $C_1$ being adjacent to $Y$, $C_2$ having a path to $Y$ through $X_2$, and $C_3$ being connected to $Y$ as $C_3 \rightarrow C_2 \rightarrow X_1 \rightarrow Y$ demonstrate that every context is non-redundant. We provide an efficient algorithm for obtaining a unique, maximal, non-redundant MPS (nr-mps, Alg. 2) of a given MPS in Appendix E [28].

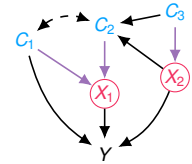

Figure 4: A non-redundant MPS

### 3.1 Non-Redundancy under Optimality

Non-redundancy of MPS (Def. 3) based on a stringent constraint imposed on $\boldsymbol{\pi}'$ is insufficient to understand, e.g., whether a context of an action would be still relevant even when $\boldsymbol{\pi} \sim \mathcal{S}$ is fully-optimized. Hence, we characterize the non-redundancy of MPS under optimality, which has practical implications to an agent adapting its suboptimal policy. Recall Fig. 3c where $X_2$ listens to $X_1$ as context. We showed that the dependence is vanished under the optimality (Fig. 3d). That is, the agent would better avoid learning $\pi(x_2|c, x_1)$ at the beginning, but optimize $\pi(x_2|c)$ instead.

**Definition 4** (Non-Redundancy under Optimality (NRO))**.** Given $\langle \mathcal{G}, Y, \mathbf{X}^\star, \mathbf{C}^\star \rangle$, an MPS $\mathcal{S}$ is said to be *non-redundant under optimality* if there exists an SCM $\mathcal{M}$ compatible with $\mathcal{G}$ such that $\mu_{\mathcal{S}}^* > \mu_{\mathcal{S}'}^*$ for every strictly subsumed MPS $\mathcal{S}' \subsetneq \mathcal{S}$, i.e., $\exists_{\mathcal{M} \sim \mathcal{G}} \forall_{\mathcal{S}' \subsetneq \mathcal{S}} (\mu_{\mathcal{S}}^* > \mu_{\mathcal{S}'}^*)$.

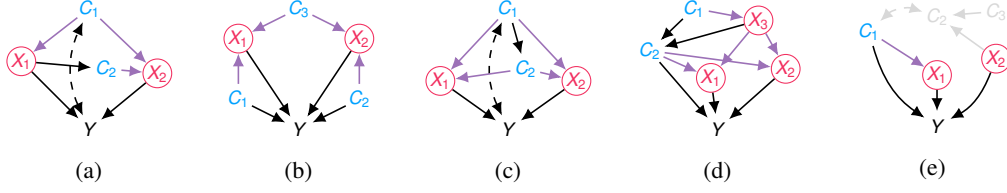

(a)          (b)          (c)          (d)          (e)

Figure 5: Causal graphs exemplifying redundancies of (a) $C_2 \to X_2$ by deterministic relationships, edges to $X_1$ and $X_2$ from (b) $C_3$, (c) $C_2$, (d) $X_3$ due to marginally or conditionally fixable contexts; (e) represents a maximal, non-redundant MPS under an optimal condition for Fig. 4.

We will investigate a criterion more general than Thm. 1—whether, for a *set* of actions $\mathbf{X}' \subseteq \mathbf{X}^\star$, a *set* of contexts $\mathbf{C}' \subsetneq \mathbf{C}_{\mathbf{X}'} \setminus \mathbf{X}'$ are relevant while taking account of deterministic relationships (Prop. 1). One approach is to characterize an opposite condition, i.e., $\mu_{\mathcal{S}}^* = \mu_{\mathcal{S}'}^*$ for $\mathcal{S}' \subsetneq \mathcal{S}$, as follows.

**Proposition 3.** *Given an MPS $\mathcal{S}$, let $\mathbf{X}' \subseteq \mathbf{X}(\mathcal{S})$ and $\mathbf{C}' \subsetneq \mathbf{C}_{\mathbf{X}'} \setminus \mathbf{X}'$ be actions and non-action contexts of interest, respectively, and let $Q' = P_{\boldsymbol{\pi} \setminus \mathbf{X}'}$. Given a mixed policy $\boldsymbol{\pi} \sim \mathcal{S}$ optimal with respect to $\mathcal{S}$, if there exist decision rules $\{\pi'(x|(\mathbf{x}' \dot{\cup} \mathbf{c}') \cap \mathbf{c}_x)\}_{X \in \mathbf{X}'}$ such that*

$$\mu_{\mathcal{S}}^* = \sum_{y,\mathbf{c}',\mathbf{x}'} y Q_{\mathbf{x}'}'(y, \mathbf{c}') \prod_{X \in \mathbf{X}'} \pi'(x|(\mathbf{x}' \dot{\cup} \mathbf{c}') \cap \mathbf{c}_x), \qquad (2)$$

*then, $\mathbf{C}_{\mathbf{X}'} \setminus (\mathbf{C}' \dot{\cup} \mathbf{X}')$ are jointly redundant to $\mathbf{X}'$ under optimality, and $\mathcal{S}' \doteq (\mathcal{S} \setminus \mathbf{X}') \cup \{\langle X, \mathbf{C}' \cap \mathbf{C}_X \rangle\}_{X \in \mathbf{X}'}$ satisfies $\mu_{\mathcal{S}}^* = \mu_{\mathcal{S}'}^*$.*

*Proof.* This follows from the definition of non-redundancy under optimality and expected reward. $\square$

To closely investigate a sufficient condition for Prop. 3, we start by discussing the implication of deterministic relationships, which characterizes an optimal policy, on the d-separation criterion. The graphical criterion handles deterministic mechanisms (i.e., *conditional intervention*) by excluding them appearing as common causes, e.g., $\leftarrow X \rightarrow$, in a trail [39]. This corresponds to adding those *implied* variables to the conditionals, in which we explicitly represent with an operation $\lceil \cdot \rceil$ for clarity. Given conditionals $\mathbf{Z}$, the implied variables with respect to $\mathbf{Z}$ is computed as follows. Initially setting $\lceil \mathbf{Z} \rceil \leftarrow \mathbf{Z}$, we update $\lceil \mathbf{Z} \rceil \leftarrow \lceil \mathbf{Z} \rceil \cup \{X \in \mathbf{X}(\mathcal{S}) \mid \mathbf{C}_X \subseteq \lceil \mathbf{Z} \rceil\}$ until it is converged. Then, given $\boldsymbol{\pi} \sim \mathcal{S}$, an optimal policy with respect to $\mathcal{S}$, a conditional independence statement $\mathbf{W} \perp\!\!\!\perp \mathbf{T} \mid \mathbf{Z}$ for $P_{\boldsymbol{\pi}}$ becomes $\mathbf{W} \perp\!\!\!\perp \mathbf{T} \mid \lceil \mathbf{Z} \rceil$ in $\mathcal{G}_{\boldsymbol{\pi}}$. Consider $C \in \mathbf{C}_X$ for some $X \in \mathbf{X}(\boldsymbol{\pi})$. The redundancy of a single context can now be expressed as $(C \perp\!\!\!\perp Y \mid \lceil \mathbf{C}_X \setminus \{C\} \rceil)_{\mathcal{H} \setminus \{X\}}$. For instance, in Fig. 5a, $C_2$ as a context of $X_2$ is independent to $Y$ given $C_1$ in a graph with $X_2$ removed since $\lceil \{C_1\} \rceil = \{C_1, X_1\}$ and $C_2 \leftarrow X_1 \rightarrow Y$ is not a valid trail anymore. Hence, $C_2$ is removable from $\mathbf{C}_{X_2}$.

Next, we illustrate contexts that unnecessarily induce correlations among actions without any implications on $Y$ (see Appendix E.1 for the derivations of the examples in Fig. 5). In Fig. 5b, both $X_1$ and $X_2$ utilize $C_3$ as their contexts. where $\mu_{\boldsymbol{\pi}} = \mathbb{E}_{c_3}[\mathbb{E}_{\boldsymbol{\pi}}[y|c_3]]$. Since there exists $c_3^* = \arg\max_{c_3 \in \mathfrak{X}_{C_3}} \mathbb{E}_{\boldsymbol{\pi}}[y|c_3]$, we can derive that $\mu_{\boldsymbol{\pi}} \leq \mathbb{E}_{\boldsymbol{\pi}}[y|c_3^*]$. Given that $c_3^*$ is merely a constant, new decision rules $\pi'(x_i|c_1) \doteq Q(x_i|c_1, c_3^*) = \pi(x_i|c_1, c_3^*)$ for $i \in \{1, 2\}$ yield the same optimal reward. A more sophisticated example is shown in Fig. 5c where a redundant context can be *fixed* conditioned on the remaining contexts. The expected reward is expressed as

$$\mu_{\boldsymbol{\pi}} = \sum_{c_1, c_2} Q(c_2|c_1)\big(\sum_{y, \mathbf{x}} y P_{\mathbf{x}}(y, c_1) \pi(x_1|c_1, c_2) \pi(x_2|c_1, c_2)\big) = \sum_{c_1, c_2} Q(c_2|c_1) \mu_{\boldsymbol{\pi}}(c_1, c_2).$$

Let $c_2^*$ be a function taking $c_1$ such that $c_2^*(c_1) = \arg\max_{c_2} \mu_{\boldsymbol{\pi}}(c_1, c_2)$ for $c_1 \in \mathfrak{X}_{C_1}$. Then,

$$\leq \sum_{c_1} \mu_{\boldsymbol{\pi}}(c_1, c_2^*(c_1)) = \sum_{y, c_1, \mathbf{x}} y P_{\mathbf{x}}(y, c_1) \pi(x_1|c_1, c_2^*(c_1)) \pi(x_2|c_1, c_2^*(c_1)).$$

By incorporating $c_2^*$ into $\boldsymbol{\pi}$, we can introduce $\boldsymbol{\pi}'$ such that $\pi(x_1|c_1, c_2^*(c_1)) \pi(x_2|c_1, c_2^*(c_1)) = \pi'(x_1|c_1) \pi'(x_2|c_1)$, satisfying Prop. 3. The variables being fixed are not necessarily conditioned on its parents (or ancestors). An example conditioning on its child is illustrated in (Fig. 5d) where we can elicit, e.g., $\pi(x_1|x_3^*(c_2), c_2) \doteq \pi'(x_1|c_2)$.

Given a general causal graph and an MPS, the aforementioned phenomena can be arbitrarily complex. We present a general criterion to test such redundancies by first proposing a lemma to obtain an intermediate expression. Let $\mathbf{V}_{\prec V}$ denote a subset of $\mathbf{V}$ preceding $V \in \mathbf{V}$ given an order $\prec$ over $\mathbf{V}$.

**Lemma 1.** *Given an MPS $\mathcal{S}$, which satisfies non-redundancy (Thm. 1), let $\mathbf{X}' \subseteq \mathbf{X}(\mathcal{S})$, actions of interest, $\mathbf{C}' \subsetneq \mathbf{C}_{\mathbf{X}'} \backslash \mathbf{X}'$. non-action contexts of interest. If there exists a subset of exogenous variables $\mathbf{U}'$ in $\mathcal{G}_{\mathcal{S}}$, a subset of endogenous variables $\mathbf{Z}$ in $\mathcal{G}_{\mathcal{S}}$ that disjoints with $\mathbf{C}' \, \dot{\cup} \, \mathbf{X}'$ and subsumes $\mathbf{C}_{\mathbf{X}'} \backslash (\mathbf{C}' \, \dot{\cup} \, \mathbf{X}')$, and an order $\prec$ over $\mathbf{V}' \doteq \mathbf{C}' \, \dot{\cup} \, \mathbf{X}' \, \dot{\cup} \, \mathbf{Z}$ such that*

1. *$(Y \perp\!\!\!\perp \boldsymbol{\pi}_{\mathbf{X}'} \mid \lceil \mathbf{X}' \, \dot{\cup} \, \mathbf{C}' \rceil)_{\mathcal{G}_{\mathcal{S}}}$,*
2. *$(C \perp\!\!\!\perp \boldsymbol{\pi}_{\mathbf{X}'_{\prec C}}, \mathbf{Z}_{\prec C}, \mathbf{U}' \mid \lceil (\mathbf{X}' \, \dot{\cup} \, \mathbf{C}')_{\prec C} \rceil)_{\mathcal{G}_{\mathcal{S}}}$ for every $C \in \mathbf{C}'$, and*
3. *$\mathbf{V}'_{\prec X}$ is disjoint with $de(X)_{\mathcal{G}_{\mathcal{S}}}$ and subsumes $pa(X)_{\mathcal{G}_{\mathcal{S}}}$ for every $X \in \mathbf{X}'$,*

*then, the expected reward for $\boldsymbol{\pi}$, a deterministic policy optimal with respect to $\mathcal{S}$, can be written as*

$$\mu_{\boldsymbol{\pi}} = \sum_{y,\mathbf{c}',\mathbf{x}'} y Q'_{\mathbf{x}'}(y, \mathbf{c}') \sum_{\mathbf{u}',\mathbf{z}} Q(\mathbf{u}') \prod_{Z \in \mathbf{Z}} Q(z|\mathbf{v}'_{\prec z}, \mathbf{u}') \prod_{X \in \mathbf{X}'} \pi(x|\mathbf{c}_x). \tag{3}$$

Lemma 1 offers a sufficient condition for obtaining the intermediate expression (Eq. (3)) for us to rewrite $\mu_{\boldsymbol{\pi}}$ as proposed in Prop. 3. The order $\prec$ dictates how the chain rule is applied in deriving the expression and what variables will appear as conditional for the probability terms. The first two conditions are relevant to separate $Q'_{\mathbf{x}'}(y, \mathbf{c}')$ from the rest. The third one is to obtain $\pi(x|\mathbf{c}_x)$ from $Q(x|\mathbf{v}'_{\prec x}, \mathbf{u}')$. We revisit Fig. 4 where we will ultimately show that, indeed $C_2$ and $C_3$ are *redundant contexts under optimality*. Given $\mathbf{C}' = \{C_1\}$ and $\mathbf{X}' = \{X_1, X_2\}$, consider $\mathbf{Z} = \{C_2, C_3\}$, $\mathbf{U}' = \emptyset$, and order $\prec = \langle C_3, C_1, X_2, C_2, X_1 \rangle$. We can derive the following expression for the expected reward (with subscripts concatenated),

$$\mu_{\mathcal{S}}^* = \textstyle\sum_{y,\mathbf{x},c_1} y Q'_{\mathbf{x}}(y|c_1) \sum_{c_{23}} Q(c_{123}, \mathbf{x}) \tag{4}$$

$$= \textstyle\sum_{y,\mathbf{x},c_1} y Q'_{\mathbf{x}}(y|c_1) \sum_{c_{23}} Q(c_3)Q(c_1|c_3)Q(x_2|c_{13})Q(c_2|c_{13}, x_2)Q(x_1|c_{123}, x_2) \tag{5}$$

$$= \textstyle\sum_{c_3} Q(c_3) \sum_{y,\mathbf{x},c_1} y Q'_{\mathbf{x}}(y, c_1) \sum_{c_2} Q(c_2|c_{13}, x_2)\pi(x_2|c_3)\pi(x_1|c_{12}). \tag{6}$$

We now provide a sufficient condition that further polishes the intermediate expression from Lemma 1 so as to represent it as the expected reward for a smaller MPS than the original one, fulfilling the condition presented in Prop. 3.

**Theorem 2.** *Let $\mathbf{U}'$, $\mathbf{Z}$, and $\prec$ satisfy Lemma 1. For $Z \in \mathbf{Z}$, let $\mathbf{V}_Z$ be a minimal subset of $\mathbf{V}'_{\prec Z} \cup \mathbf{U}'$ such that $Q(Z \mid \mathbf{V}_Z) = Q(Z \mid \mathbf{V}'_{\prec Z}, \mathbf{U}')$. We define $\mathsf{fix}(\mathbf{T})$ with respect to $\{\langle Z, \mathbf{V}_Z \rangle\}_{Z \in \mathbf{z}}$, that is, with $\hat{\mathbf{T}} \doteq \lceil \mathbf{T} \rceil \cup \{Z \in \mathbf{Z} \mid \mathbf{V}_Z \backslash \mathbf{U}' \subseteq \lceil \mathbf{T} \rceil\}$, $\mathsf{fixed}(\mathbf{T})$ is $\mathbf{T}$ if $\mathbf{T} = \hat{\mathbf{T}}$ and $\mathsf{fixed}(\hat{\mathbf{T}})$, otherwise. If $\mathsf{fixed}(\mathbf{C}_X \backslash \mathbf{Z}) \supseteq \mathbf{C}_X$ for $X \in \mathbf{X}'$, then, $\mathcal{S}' \doteq (\mathcal{S} \backslash \mathbf{X}') \cup \{\langle X, \mathbf{C}_X \backslash \mathbf{Z} \rangle\}_{X \in \mathbf{X}'}$ satisfies $\mu_{\mathcal{S}'}^* = \mu_{\mathcal{S}}^*$.*

Thm. 2 provides a condition where Eq. (3) can be transformed to $\mu_{\mathcal{S}'}^*$. To do so, it examines whether terms $Q(z|\mathbf{v}_z)$ can be removed by fixing $Z$ to $z^*$ conditional on $\mathbf{v}_z$ in connection with the context to be removed. That is,

$$\mu_{\mathcal{S}}^* = \overbrace{\sum_{\mathbf{u}'} Q(\mathbf{u}')}^{\text{marginally fixable}} \sum_{y,\mathbf{c}',\mathbf{x}'} y \underbrace{Q'_{\mathbf{x}'}(y, \mathbf{c}')}_{\text{irrelevant to } \mathbf{Z}} \sum_{\mathbf{z}} \underbrace{\prod_{Z \in \mathbf{Z}}}_{\text{defines dependency}} \overbrace{Q(z|\mathbf{v}_z)}^{\text{to fix conditionally}} \prod_{X \in \mathbf{X}'} \underbrace{\pi(x|\overbrace{\mathbf{c}_x \backslash \mathbf{z}}^{\text{given}}, \overbrace{\mathbf{c}_x \cap \mathbf{z}}^{\text{to infer}})}_{\text{to be } \pi'(x|\mathbf{c}_x \backslash \mathbf{z})}, \tag{7}$$

We explain the theorem by deriving further from Eq. (6). $C_3$ can be fixed to a constant $c_3^*$ so that,

$$\leq \textstyle\sum_{y,\mathbf{x},c_1} y Q'_{\mathbf{x}}(y, c_1) \sum_{c_2} Q(c_2|c_1, c_3^*, x_2)\pi(x_2|c_3^*)\pi(x_1|c_1, c_2). \tag{8}$$

There exists $x_2^* \in \mathfrak{X}_{X_2}$ where we can replace $\pi(x_2|c_3^*)$ with $\pi'(x_2)$ such that $\pi'(x_2^*) = 1$.

$$\leq \textstyle\sum_{y,\mathbf{x},c_1} y Q'_{\mathbf{x}}(y, c_1) \sum_{c_2} Q(c_2|c_1, c_3^*, x_2^*)\pi'(x_2)\pi(x_1|c_1, c_2). \tag{9}$$

These steps first correspond to checking $\mathsf{fixed}(\emptyset) = \{C_3, X_2\}$ and, then, safely replacing the decision rule for $X_2$ by eliminating $C_3$ from its context since $\mathsf{fixed}(\mathbf{C}_{X_2} \backslash \mathbf{Z}) \supseteq \mathbf{C}_{X_2} = \{C_3\}$. Next, the optimal $c_2$ is determined with respect to $c_1$, i.e., $Q(c_2|c_1, c_3^*, x_2^*)$, where we can replace $\pi(x_1|c_1, c_2^*(c_1))$ by $\pi'(x_1|c_1)$,

$$= \textstyle\sum_{c_1,c_2} Q(c_2|c_1, c_3^*, x_2^*) \sum_{y,\mathbf{x}} y Q'_{\mathbf{x}}(y, c_1)\pi'(x_2)\pi(x_1|c_1, c_2) \tag{10}$$

$$\leq \textstyle\sum_{y,\mathbf{x},c_1} y Q'_{\mathbf{x}}(y, c_1)\pi'(x_2)\pi(x_1|c_1, c_2^*(c_1)) \tag{11}$$

$$= \textstyle\sum_{y,\mathbf{x},c_1} y Q'_{\mathbf{x}}(y, c_1)\pi'(x_1|c_1)\pi'(x_2) = \mu_{\mathcal{S}'}^*. \tag{12}$$

These steps correspond to checking $\mathsf{fixed}(\mathbf{C}_{X_1} \backslash \mathbf{Z}) = \mathsf{fixed}(\{C_1\}) = \{C_1, C_3, X_2, C_2\} \supseteq \{C_1, C_2\}$ for $X_1$. Since $\mu_{\mathcal{S}'}^* \leq \mu_{\mathcal{S}}^*$ by the existence of $\boldsymbol{\pi} \in \mathcal{S}$ that can emulate $\boldsymbol{\pi}' \in \mathcal{S}'$, and $\mu_{\mathcal{S}'}^* \geq \mu_{\mathcal{S}}^*$ by the derivation (Eq. (12)), we can conclude that $\mu_{\mathcal{S}'}^* = \mu_{\mathcal{S}}^*$. As a consequence, MPS $\mathcal{S}$ is *not non-redundant under optimality* due to the ineffective contexts $\{C_2, C_3\}$ with respect to $\{X_1, X_2\}$.

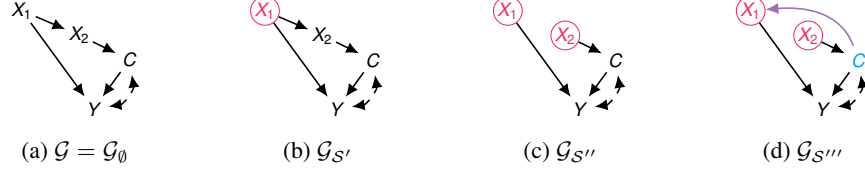

$$\text{(a) } \mathcal{G} = \mathcal{G}_\emptyset \qquad \text{(b) } \mathcal{G}_{\mathcal{S}'} \qquad \text{(c) } \mathcal{G}_{\mathcal{S}''} \qquad \text{(d) } \mathcal{G}_{\mathcal{S}'''}$$

Figure 6: A causal graph $\mathcal{G}$ (a) and its induced graphs (a,b,c,d) where the mixed policy scope on the right is better than or equal to the one on the left with respect to their optimal rewards.

## 4 A Partial Order over Mixed Policies and Possible-Optimality

Equipped with the notion of non-redundancy under optimality (NRO, Def. 4), an agent can more efficiently optimize its policy than relying on generic non-redundancy (Def. 3). Yet, an important question is whether an MPS is worth to explore for an agent to converge to an optimal policy. Consider for an instance, see Figs. 6a to 6d which represent various NRO MPSes. However, even without interacting with an environment, we can claim $\mu \leq \mu^*_{\mathcal{S}'} \leq \mu^*_{\mathcal{S}''} \leq \mu^*_{\mathcal{S}'''}$, that is, the next MPS is better than or equal to (simply *better* or *improved* hereinafter) the one regarding their optimal expected rewards in *any* model: First, $\mu \leq \mu^*_{\mathcal{S}'}$ since there exists an optimal $X_1$ value, $x_1^*$; Next, $\mu^*_{\mathcal{S}'} \leq \mu^*_{\mathcal{S}''}$, there exists an optimal $X_2$ value, and can be determined without conditional on $X_1$, which is implied; Finally, $\mu^*_{\mathcal{S}''} \leq \mu^*_{\mathcal{S}'''}$ since $X_1$ can better behave by taking an effective context $C$ into account. Therefore, the agent can only optimize parameters involving $\mathcal{S}'''$ (Fig. 6d) to obain an effective policy. Against this background, we characterize such a partial order over the space of MPSes with respect to their maximum expected rewards achievable: when one MPS is better than the other. To begin a formal discussion, we introduce *possible-optimality* of MPS.

**Definition 5** (Possibly-Optimal MPS)**.** Given $\langle \mathcal{G}, \mathbf{X}^\star, \mathbf{C}^\star, Y \rangle$, let $\mathbb{S}$ be a set of NRO MPSes. An MPS $\mathcal{S} \in \mathbb{S}$ is said to be *possibly-optimal* if there exists $\mathcal{M} \sim \mathcal{G}$ such that $\mu^*_{\mathcal{S}} > \max_{\mathcal{S}' \in \mathbb{S} \setminus \{\mathcal{S}\}} \mu^*_{\mathcal{S}'}$.

In the partial order sense, POMPSes are the maximal elements among NRO MPSes. To study the partial order, we present two operations which take an MPS and return an improved MPS: (i) adding observations for existing actions and (ii) adding new interventions. These two operations offer sufficient conditions for identifying non-POMPSes.

**Proposition 4.** *Given an MPS $\mathcal{S}$ and $X \in \mathbf{X}(\mathcal{S})$, adding $C \in \mathbf{C}^\star \setminus \{X\}$ as a context of $X$, resulting $\mathcal{S}' = (\mathcal{S} \setminus \{X\}) \cup \{\langle X, \mathbf{C}_X \cup \{C\} \rangle\}$ improves $\mathcal{S}$ if $C \notin de(X)_{\mathcal{G}_{\mathcal{S}}}$ and $C \perp\!\!\!\perp Y \mid \lceil \mathbf{C}_X \rceil$ in $\mathcal{H} \setminus \{X\}$.*

This proposition is straightforward. Note however that the resulting MPS may not be NRO as an added observation can cancel out the relevance of the existing contexts, e.g., Prop. 2 can be viewed as adding observations and removing now irrelevant observations. Further, any set of observations that can be added to a set of actions to improve an MPS can also simply be added sequentially.

**Adding new interventions** Intervention replaces the natural mechanism for $X \in \mathbf{X}^\star$ with an artificial one $\pi(x|\mathbf{z})$. To guarantee that the alternative one can perform at least as good as the natural one, we should understand what information $X$ originally takes and whether the new contexts $\mathbf{Z}$ carry information tantamount to the original one. If every parent of $X \in \mathbf{X}^\star$ is contextualizable (e.g., no UC), the problem becomes trivial (e.g., Markovian). Otherwise, we examine the existence of a back-door path.[3] Let $Q = P_{\boldsymbol{\pi}}$ and $\mathcal{H} = \mathcal{G}_{\boldsymbol{\pi}}$ for some $\mathcal{S} \sim^{-1} \boldsymbol{\pi}$. Given $X \in \mathbf{X}^\star \setminus \mathbf{X}(\boldsymbol{\pi})$ and $\mathbf{Z} \subseteq \mathbf{C}^\star \setminus \{X\}$, if (i) $(Y \perp\!\!\!\perp X \mid \lceil \mathbf{Z} \rceil)_{\mathcal{H}_{\underline{X}}}$ and (ii) $X \notin an(\mathbf{Z})_{\mathcal{H}}$, then

$$\mu_{\boldsymbol{\pi}} = \sum_{y,x,\mathbf{z}} y Q(y|x,\mathbf{z})Q(x|\mathbf{z})Q(\mathbf{z}) \overset{\text{(i)}}{=} \sum_{y,x,\mathbf{z}} y Q'_x(y|\mathbf{z})Q(x|\mathbf{z})Q(\mathbf{z})$$
$$\overset{\text{(ii)}}{=} \sum_{y,x,\mathbf{z}} y Q'_x(y|\mathbf{z})Q(x|\mathbf{z})Q'_x(\mathbf{z}) \doteq \sum_{y,x,\mathbf{z}} y Q'_x(y,\mathbf{z})\pi'(x|\mathbf{z}) \doteq \mu_{\boldsymbol{\pi}'},$$

for some $\pi'$. Since $\pi'$ can be optimized, $\mu^*_{\mathcal{S}} \leq \mu^*_{\mathcal{S} \cup \{\langle X, \mathbf{Z} \rangle\}}$. However, naively generalizing the criterion to handle a set of interventions is insufficient. Consider Fig. 7a, an observational policy where $\mathbf{X} = \mathbf{X}^\star$ and $\mathbf{C} = \mathbf{C}^\star$. Based on the aforementioned criteria, $X_1$ and $X_2$ shall not be intervened simultaneously (by replacing $X$ to $\mathbf{X}$): $C_2$ cannot be used as $\mathbf{Z}$ since $X_1 \in an(C_2)_{\mathcal{G}}$; $X_2 \leftrightarrow C_2 \to Y$ is an open back-door path. We propose a solution for adding interventions simultaneously, powered by Thm. 2.

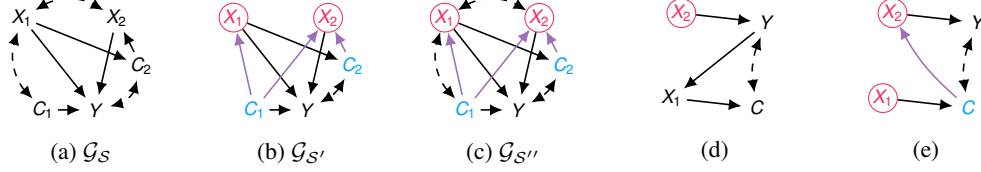

Figure 7: (a) a given MPS to construct (c) an improved MPS through (b) an intermediate, invalid MPS. (d,e) demonstrate the use of post-reward interventions to improve a given MPS.

**Theorem 3.** *Given an MPS $\mathcal{S}$, let $\mathcal{S}' \neq \mathcal{S}$ be an MPS with $\mathbf{X}(\mathcal{S}) \subseteq \mathbf{X}(\mathcal{S}')$ such that $\mathcal{H}''$ the union of induced graphs $\mathcal{G}_\mathcal{S} \cup \mathcal{G}_{\mathcal{S}'}$ is acyclic. Let $\mathbf{X}'$ be actions that the MPSes disagree on, i.e., $(\mathbf{X}(\mathcal{S}') \backslash \mathbf{X}(\mathcal{S})) \cup \{X \in \mathbf{X}(\mathcal{S}) \mid \mathbf{C}'_X \neq \mathbf{C}_X\}$, and (invalid) MPS $\mathcal{S}'' \doteq \{\langle X, pa(X)_{\mathcal{H}''} \cup \mathbf{U}_X\rangle\}_{X \in \mathbf{X}'}$. $\mu^*_{\mathcal{S}''} = \mu^*_{\mathcal{S}'}$ can be elicited by Thm. 2, then, $\mu^*_\mathcal{S} \leq \mu^*_{\mathcal{S}'}$.*

Given an MPS $\mathcal{S}$, an intermediate MPS is constructed adding new contexts to a subset of $\mathbf{X}^\star$ while assuming that any non-contextualizable variables can be used as contexts. Consider comparing $\mathcal{S} = \emptyset$ (Fig. 7a) and $\mathcal{S}'$ (Fig. 7b) where we employ an intermediate representation $\mathcal{S}''$ (Fig. 7c) to ultimately inspect $\mu^*_\mathcal{S} \leq \mu^*_{\mathcal{S}'}$. Thm. 2 is applicable with $\mathbf{U}' = \emptyset$ and $\prec = \langle C_1, \mathbf{U}_{X_1}, \mathbf{U}_{X_2}, X_1, C_2, X_2\rangle$ to demonstrate $\mu^*_{\mathcal{S}''} = \mu^*_{\mathcal{S}'}$. Since $\mu^*_\mathcal{S} \leq \mu^*_{\mathcal{S}''}$, we can elicit $\mu^*_\mathcal{S} \leq \mu^*_{\mathcal{S}'}$, confirming that $\mathcal{S}$ is not a POMPS. By allowing $\mathbf{X}'$ to intersect with $\mathbf{X}(\mathcal{S})$, the theorem not only adds new inventions but also can replace the contexts of existing interventions.

**Refining the space of MPSes** Equipped with the characterizations, we can refine the space of MPSes, hence, the space of mixed policies, by filtering out MPSes that are either redundant or dominated by other MPS, eliciting a superset of POMPSes in a given setting. This can be achieved in a brute-force manner by enumerating all MPSes, and examining whether any of Thm. 2, Thm. 3, and Prop. 4 is applicable. One of barriers to design a more principled approach (e.g., dynamic programming [16]) to obtaining POMPSes (or a superset of) is that contexts are interleaved in both terms in Eq. (1) representing a reward mechanism and a policy.

Nevertheless, we investigate simplifying a mixed policy setting while preserving its POMPSes. First, one may think that the descendants of $Y$ can be ignored since neither intervening action variables among them changes the reward nor observing contextualizable variables among them is feasible. Surprisingly, Fig. 7d, where $X_1$ and $C$ take place *after* the reward is evaluated, remarkably demonstrates the opposite. With $X_1$ intervened on, $C$ can become a context for $X_2$ (Fig. 7e) without inducing a cycle. This implies that contexts in the descendants of the reward becomes usable if interventions can break the ancestral relationships. Second, $X \in \mathbf{X}^\star$ that cannot affect $\mathbf{C}^\star$ or $Y$ is not intervene-worthy — if $de(X)_\mathcal{G} \cap (\mathbf{C}^\star \cup \{Y\}) = \emptyset$, there exists no MPS that makes $X \in an(\mathbf{C}^\star \cup \{Y\})_\mathcal{G}$, and, thus, $X$ can be excluded from $\mathbf{X}^\star$.

**Proposition 5.** *Given $\langle\mathcal{G}, Y, \mathbf{X}^\star, \mathbf{C}^\star\rangle$, let $\mathbf{X}' \doteq \{X \in \mathbf{X}^\star \mid de(X)_\mathcal{G} \cap (\mathbf{C}^\star \cup \{Y\}) \neq \emptyset\}$, $\mathbf{X}'' \doteq de(Y)_{\mathcal{G}_{\overline{\mathbf{X}'}}} \cap \mathbf{X}'$, and $\mathbf{Z} \doteq de(Y)_{\mathcal{G}_{\overline{\mathbf{X}''}}}$. The POMPSes for $\langle\mathcal{G}, Y, \mathbf{X}^\star, \mathbf{C}^\star\rangle$ are the same as those for $\langle\mathcal{G}\backslash\mathbf{Z}, Y, \mathbf{X}', \mathbf{C}^\star\backslash\mathbf{Z}\backslash\mathbf{X}''\rangle$.*

## 5 Conclusions

In this paper, we studied the space of mixed policies that emerges through the empowerment of an agent to determine the mode it will interact with the environment — i.e., which variables to intervene on and which contexts it decides to look into. Facing new challenges to optimize this new mode of interaction, which has many additional degrees of freedom, we studied the topological structure induced by the different mixed policies, which could in turn be leveraged to determine partial orders across the policy space w.r.t. the maximum expected rewards achievable. As a practical result, we provided a general characterization of the space of mixed policies with respect to properties that allow the agent to detect inefficient and suboptimal strategies. One of the surprising implications of this characterization provided here is that agents following a more standard approach (i.e., intervening on all intervenable variables and observing all available contexts) may be hurting themselves, and may never be able to achieve an optimal performance regardless of the number of interactions performed.

## Broader Impact

Our work investigates the efficiency and effectiveness of AI agents to explore the environments and ultimately achieve optimality. Our results provide a tool for AI engineers and researchers to identify where the inefficiency of a policy may be coming from, including potentially unintended side effects. Further, the characterization provided in this work can suggest how systemic improvements are possible given non-parametric causal understanding of the underlying systems. The very topic of our paper about efficiency and effectiveness has been studied for several decades in diverse fields: bandits, reinforcement learning, design of experiments, etc. Hence, it is not difficult to imagine that our work will share the common problems with other automated decision making tools and methods such as (i) the system optimized based on an ill-defined reward may harm 'unknown unknowns' (e.g., increasing the revenue of alcoholic beverage companies based on targeted advertising over recovering alcoholics if their health is not properly modeled) or (ii) the optimization can be impossible due to the participants of adversarial players (e.g., rewarding the number of software bugs fixed makes software engineers to create more bugs to fix, see Goodhart's law). Mitigating the first kind of risks will require deploying proper countermeasure through, e.g., regulations by governments. The second kind of risks (errors or failures) can be detected through examining possible changes of underlying mechanisms (i.e., anomaly detection). However, the current work does not consider multiple adversarial participants (e.g., game-theoretic settings), which is a subject of future research.

## Acknowledgments and Disclosure of Funding

This research is supported in parts by grants from NSF (IIS-1704352 and IIS-1750807 (CAREER)).

## Footnotes

[1]This condition was leveraged in the atomic interventions case to establish minimality [16, 17]; see also [40].

[2]The relevance of contextual information has been discussed in the influence diagrams literature [41, 20]. More recently, this condition was used in the case of singleton decisions (i.e., $|\mathbf{X}(\mathcal{S})| = 1$), see [42, 43].

[3][16, 17] studied 'possibly-optimal' atomic interventions ($\mathbf{C}^\star = \emptyset$) where their conclusions can be essentially reduced to finding actions with no back-door path to $Y$ while varying the strengths of UCs.

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
