[Reviews · NeurIPS 2020]

Review 1

Summary and Contributions: After rebuttal: I thank the authors for taking the time to carefully respond to my comments. I've updated my score to reflect the authors' comments. In carrying out their revisions, I want to stress that the authors should pay attention to points about "simplicity" and references to the RL literature. "Simplicity" -- The authors seem to be indicating they're using a statistical notion (i.e. fewer parameters to estimate). This should be formalized more clearly and also the authors might consider making concrete how the notions of simplicity play out in the context of a hypothetical data application (e.g. ascribe a story to the graphs/interventions). RL references: it seems that multiple reviewers raised points about making connections to the RL literature stronger. This is not my primary area of research so I'm afraid I can't provide constructive recommendations but the authors should try their best to expand the number of references so as to bring this work closer to the broader NeurIPS audience. Definition 1/Acyclicity -- this was mostly a misreading on my part, with the stress of the relevant sentence made somewhat confusing by the comma between "such that X \in ... \{X\} (comma)" and "and G_S is acyclic, where...". A clearer rewriting might be "S is defined as a collection of pairs <X, C_X> such that (1) X \in ... \{X\}, and (2) G_S is acyclic, where...". This would help clarify that both (1) and (2) are part of the "such that" clause. ------------------------------------------------- This paper describes a formalism for characterizing the optimality of policy interventions in causal DAGs. The authors give procedures for determining when seemingly distinct policies are either equivalent or when one is subsumed by the other. They also provide a framework for determining which of a collection of candidate interventions is optimal with respect to a user-defined outcome.

Strengths: The paper is well motivated. The authors provide several well-developed examples to describe each step of the development of the formalism. I read the proofs in modest detail and, to the best that I can tell, they are sound. Significance and novelty: Many of the ideas involved in the definitions of policies and the mechanics for adopting them appear to be minimally novel. The main novelty appears to be in the identification of redundancy and comparisons of optimality. I have some interpretational questions (below) that make me question the degree to which the redundancy contribution is a high impact one. It does appear to be novel for this setting, however. With respect to optimality, to my knowledge these results are novel from a causal perspective but I cannot comment on novelty from the reinforcement learning angle.

Weaknesses: The primary concerns I have are hinted at above. These are with respect to redundancy and optimality. Redundancy: I question the efficacy of the redundancy-of-policies results. A good example of language that raises this concern for me is at the bottom of page 4. The authors say "By noting that the policy relative to Fig. 3d can achieve optimality, while being simpler than the one relative to Fig. 3c, we investigate how to capture non-redundancy within MPSes". This sentence, stressing "simplicity", seems to be the primary motivation for a large chunk of the remainder of the paper. What's odd, though, is that the notion of simplicity isn't really defined to this point, either formally or informally. While it is intuitively clear why the two MPSs capture the same policy intervention dynamics, it is not clear why 3d is preferable relative to 3c. It might be the case that the analyst wishes to explicitly study the graph in Fig 3c because this fits some sort of intuitive notion of what the "post-intervention" dynamics should look like, while 3d might not. The authors should clarify this motivation in more detail in a reworked draft. In addition to generally increasing the clarity on this issue, the authors should be sure to clarify the definition of simplicity. For instance, is it a statistical notion? A graphical complexity notion? I cannot adequately comment on the novelty of the optimality results from a reinforcement learning perspective as this is not my subject area. That said, it seems, based primarily on tracking the citations in the paper, that there is minimal (if any) effort to connect the results in this paper to those known from the broader ML literature. The authors cite several RL works in the introduction and motivation but do not place their results in context once they've been detailed in the main body of the paper. This makes it hard to judge the degree to which these results are novel or useful in the context of the broader literature. Additional concerns are highlighted in the comment section below.

Correctness: The proofs surrounding the main results appear to be largely sound. There are some issues (possibly merely typos) with the setup surrounding these results that jeopardize the overall correctness if not addressed. These are highlighted in the detailed comments section below.

Clarity: For the most part the paper is well written, aside from comments made elsewhere in this review that ask the authors for specific clarification.

Relation to Prior Work: See above comments regarding optimality. The authors should make more clear how their results compare to existing results in RL. Additionally, there are a few cases where additional citations of similar work on policy interventions in causal inference should be considered. These are: At the end of page 1 for the 22-26 citation block, 1) "A new approach to causal inference in mortality studies with a sustained exposure period—application to control of the healthy worker survivor effect" Robins 1986, which lays a lot of the ground work for the sequential decision-making framework in causal models and pairs nicely with citation 22, and 2) "Identification of Personalized Effects Associated with Causal Pathways" Shpitser and Sherman 18, which provides completeness results for Tian's 2008 paper (citation 26). Additionally, many of the ideas and language the authors use with respect to specifying which variables a policy "listens to" reflect similar ideas discussed in "General Identification of Dynamic Treatment Regimes Under Interference" Sherman, Arbour, and Shpitser 2020. While this paper primarily focused on identification issues, the formalisms and notions of which variables are "listenable" and what implications that has for inference are similar and so the connection should be noted as a parallel work.

Reproducibility: Yes

Additional Feedback: Identification: The authors don't really address the fact that the effects of some policies may not be identifiable. In these cases do redundancy results still hold? As an example, in Fig. 1, Y(X2) is not identifiable due to the bow-arc. By the result in SS18 (see above), this means the effect of a policy intervention on X2 is not identifiable. Are there unstated assumptions that enable the authors to get around this issue? DAGs vs. Mixed Graphs -- second paragraph of Preliminaries starts with "Graphically, each SCM (..) is associated with a directed acyclic graph (..) where (...) (ii) X <-> Y (represents) the existence of an unobserved confounder". If there are <-> edges then the graphs used here are not DAGs (though the underlying causal model is a DAG and perhaps the intention is to study graphs that are equivalence classes of DAGs?). This issue also comes up in the statement of Theorem 3. The authors say "... where the unobserved confounders adjacent to X' are made explicit". This doesn't have a formal meaning I am aware of. Is the idea to replace <-> edges with a single unobserved variable with edges that point to the two endpoints of the <-> edge? This would mean effectively selecting one member of the equivalence class. Does this result then hold for the entire equivalence class? This is not adequately formal. Use of the word 'observe' -- at the beginning of section 2 the authors say "a causal understanding of the world gives rise to a rich spectrum of policies with different scopes, allowing agents to choose (...) which variables to intervene and to observe". Isn't the graph taken as given here and so what's observed/unobserved (e.g. observed vs. latent) is also a given? Do the authors mean "which variables to 'listen' to"? This language is used in various places throughout (e.g. also at top of section 3) and should be clarified. Acyclicity and further intersection with SAS20 (see above) -- definition 1 specifies that G_S is acyclic. Is this a condition on the mixed policy scope? I.e. A pair is not an MPS if G_S is cyclic? This is a bit unclear from the definition present and should perhaps be made formal. This also gives rise to further connection to Sherman, Arbour, Shpitser 2020. In that paper the authors, though considering a more general class of graphs, defined policy interventions in a way such that they would not induce cycles and proved that the post-intervention graphs were acyclic. It seems in this case that either a similar result is necessary for MPSs or that the result is a corollary of the result in SAS20 (likely the former since MPSs have a similar, but not identical, formalism). POMPSes -- the authors might consider renaming this construct since the acronym reads like the English pejorative term "pompus".


Review 2

Summary and Contributions: The authors feedback has been more explicit about the meaning of some of the important terms in their paper (e.g "simplicity", that the graphs are ADMGs, the role of standard latent projections, etc.). The terminology and basic assumptions should be made more explicit to the reinforcement learning community, but I am confident that the changes required to do so are relatively minor and would be easy to make. My original evaluation was positive, and remains so after the authors feedback. -------------------------------------------------------------------------------------------- This paper describes how to combine reinforcement learning and causal inference in order to search for optimal policies more effectively. Given an input graph representing the causal relations between context variables, action variables, and a goal variable, they describe how to detect when some policies are dominated by other policies in terms of the maximum achievable expected value for the goal, and when adding more actions and contexts to the space to explore will not improve the maximum achievable expected value of the goal.

Strengths: The problem addressed in this paper is of interest to the NeurlIPS community, in that it draws interesting connections between reinforcement learning and causal inference. It is not clear at this stage how useful in practice it would be, but it is thoeretically interesting, and at least worth developing further.

Weaknesses: There is no complexity analysis of the algorithms for determining redundancy of contexts and actions, and no complexity analysis of the algorithms for determining when a policy is dominated by other policies (although the supplementary material does mention that some of these algorithms are polynomial). There are some simple examples of this in the supplementary material showing that employing the algorithms lead to convergence to an optimal strategy with many fewer steps than e.g. brute force searching, but they do not take into account the cost of calculating which policies are redundant or dominated. If search strategies are to be made more effective by eliminating some policies which employ these algorithms (as opposed to e.g. just brute force checking all of them) this would be more persuasive in showing that employing them would be valuable. In addition, while the current version of the paper does illustrate that some simple search strategies that are intuitively plausible actually can prevent one from finding the optimal policy, it is not clear how common these would be (given that their example is rather contrived). If they could find space to even briefly mentioned the results of their simulation in the supplementary material, that would be helpful to the reader. Another limitation to the practical usefulness of the approach in this paper is that it requires a known causal graph for the contexts, actions, and outcomes. It is not clear how often this would be available in practice.

Correctness: The paper discusses a method of extending d-separation that handles the extra conditional independencies which may be entailed by determinism among sets of variables. Unlike d-separation, it is known that the method they describe is not complete however. For example, if a child determines the value of a parent, then conditioning on the child can entail an independence even if it is parent that is blocking the path. I don't know whether those extra independencies matter to their conclusions in any way.

Clarity: The paper is quite well written. They illustrate most of their definitions with simple examples, which helps clarify them a lot.

Relation to Prior Work: The paper clearly discussed how this work differs from previous contributions.

Reproducibility: Yes

Additional Feedback:


Review 3

Summary and Contributions: EDIT after rebuttal: Overall, I'm satisfied with the authors rebuttal and still positively inclined to the paper. The paper presents an extension of the structural bandit line of research [16,17] in which there is a structured space of possible actions and there might be confounding, to the case in which we also want to minimize which variables we observe. The authors propose a graphical criterion that detects redundant policies and sufficient conditions for policies to dominate each other.

Strengths: The paper is theoretically strong and it provides an interesting and novel generalization of a potentially very promising line of work (structural bandits) with wide-ranging implications for the bandits community.

Weaknesses: In my opinion, the paper is very good, but it still has two main weaknesses that slightly hinder its significance: - the accessibility for a more general ML audience and the fact that a lot of the content (including some evaluation) is in the Appendix - not enough emphasis on the strong assumptions, e.g. knowing the causal graph On the other hand, I can understand the first point is difficult to fix, also given the paper would probably fit better a journal in order to be properly self contained.

Correctness: I haven't checked the proofs in too much detail, but as far as I could see the paper seems correct.

Clarity: The paper is generally well-written for a causality audience, but probably not very clear to the general NeurIPS attendance. A lot of the useful content is in the Appendix, for example the description of possible actions in the first example, or some empirical evaluation.

Relation to Prior Work: The relation to previous work is very clear and well-explained, since the paper is a direct extension of the structured bandits framework [16,17].

Reproducibility: Yes

Additional Feedback: I would probably move an even bigger part of L87 to L114 (Preliminaries) to Appendix B, and create some more space for explaining the running example (now in Appendix A). L170 etc. listening seems a bit a weird choice of verb, would "X2 is determined by X1" (or depends on) maybe be a better choice?


Review 4

Summary and Contributions: This work gives a theoretical exploration of, given a world described by a causal graph, how to reasonably define subsets of interventional and context variables that can theoretically achieve optimal performance without redundancy. This requires formally introducing several concepts about causal graphs and interventions and contexts. I should say at the outset that I am not an expert in this area. I have done empirical RL work and have read about causal graphs / causal theory (e.g. several papers from Judea Pearl), but have not used it actively in my work.

Strengths: This work clearly defines and introduces concepts with solid theoretical grounding. Abstractly, the work is trying to answer questions about the structure of models to fit: what variables are inputs and what variables are output if you want optimal behavior of an agent? This is an important broad question. Given a world described by a causal graph, this paper provides theoretically sound answers (see caveats below). This solid theoretical exploration should provide solid concepts for future work to build on.

Weaknesses: For the specific area of causal reasoning and AI, I don’t know this sub-field well enough to be able to comment on the importance of this work. I think this work is unlikely to have short term impact on practice. The use of causal graphs to describe environments is not that common in general. However, even if they were, I can not tell whether the implications of the theory described produces conclusions that would be surprising / that would change practice. The “one surprising result” in the abstract is not well characterized. What kinds of graphs produce this surprising result? This is not well explained in the paper and makes it very difficult to understand if that result has any practical significance. There are some very nice examples (like Fig 7e) about surprising structure. Lines 69-71 give an informal definition of optimality that, in my reading, conflicts with the later definition. I read this as “optimal means there does not exist a policy strictly better in every world”. But the later more formal definition (Def 5), I think, allows there to be some worlds where there is no *strictly* better policy, but the policy is question is *equal* to the optimal. Plus my reading of the informal definition produces a very strange meaning for optimal because you can always consider the world where Y ends up independent from all other variables (even though there are edges in the graph, a particular realization of F could ignore a particular input) so all policies have equivalent values. (Though I am not an expert, so my apologies if I have misunderstood).

Correctness: I did not verify any of the proofs in the supplementary material so I can not comment directly on the correctness of those. Everything in the structure of the claims seems sound.

Clarity: For a non-expert in this area, I found the work to be extremely well written. The notation was clear and well chosen and allowed me to understand the reasoning and claims of the work with a reasonable amount of effort. Thank you so much for the time you clearly devoted to this! The one exception is Theorem 2 which I was not able to make sense of. This could entirely be my own failing, so if other reviewers had no issues I wouldn’t worry about it.

Relation to Prior Work: I am not familiar enough with this field to comment.

Reproducibility: Yes

Additional Feedback:

[Author Response · NeurIPS 2020]

¹ We are glad to see that the reviewers found our work interesting, relevant, and sound. We also appreciate the feedback
² provided, which will be incorporated to improve the manuscript. Below, we address each reviewer's comments.

³ **Reviewer #1** [**A**. *Term 'Simplicity'*]: The term simplicity in this context is about having fewer actions and contexts
⁴ while guaranteeing the same optimal performance. Without simplicity, additional parameters that are irrelevant will
⁵ need to be estimated during the exploration stage, and will possibly slow down the whole learning process. To ground
⁶ this discussion, we contrast the MPSes of Fig. 3c and Fig. 3d. In particular, the MPS in Fig. 3d is $\pi(X_1|C)\pi(X_2|C)$
⁷ while the one in Fig. 3c is $\pi(X_1|C)\pi(X_2|C,X_1)$. While both MPSes coincide in terms of $\pi(X_1|C)$, the second
⁸ component of MPS 3d — $\pi(X_2|C)$ — is "simpler" (i.e., lower dimensional) than that of MPS 3c, $\pi(X_2|C,X_1)$. We'll
⁹ try to improve the discussion in the paper, thanks. [**B**. *Put in the context of RL*]: This work focuses on policies over
¹⁰ a general causal graph, which traditional RL framework lacks. Actions and contexts are typically fixed a priori in
¹¹ RL. We tried to emphasize such differences through the introductory example. For instance, given a traditional MDP
¹² structure, $\{\pi(x_t|s_t)\}_t$ is considered the only POMPS. [**C**. *References*]: Thanks for the suggestions, they appear to be
¹³ interesting and relevant, will check. [**D**. *Identifiaiblity issues*]: The focus of the paper was on characterizing mixed
¹⁴ policies given that the agent can act (i.e., online learning). In other words, the focus is not in doing off-policy evaluation
¹⁵ and using the data collected by another agent, which is when identifiability comes into play. [**E**. *Term 'DAG' and*
¹⁶ *'Mixed Graphs'*]: We meant by a 'DAG with latent variables' (also called a causal diagram), which is represented
¹⁷ as an ADMG under latent projection. We follow Pearl's notation, but will add a clarification note about it, thanks.
¹⁸ [**F**. *Explicitizing bidirected edges to UCs in Thm. 3*]: Thm. 3 makes use of a specific instantiation of the unobserved
¹⁹ confounders for the sake of reduction. However, it holds true irrespective of such instantiation. In fact, one can employ
²⁰ $\mathbf{u}_X$ (which also includes variable-specific exogenous variables that can be effectively ignorable, see line 94) for $X \in \mathbf{X}'$.
²¹ For concreteness, see Fig. 7b, where the two bidirected edges are mapped to $\{U_1, U_2\}$. Whenever they appear in the
²² derivation, $\mathbf{U}_{X_1} = \{U_1, U_2\}$ can be used, $\mathbf{U}_{X_2} = \{U_1\}$, or their union, without relying on a specific instantiation.
²³ Having said that, that's a good point, we will try to make it more direct and explicit in the manuscript. [**G**. *Term*
²⁴ *'Observe' and 'Listens to'*]: The terms are used to describe 'being used as a context' (i.e., contextualized). There are
²⁵ multiple similar terms (observe, listen, see, and contextualize). We borrow the terminology from Pearl but will make
²⁶ their meanings clearer to avoid any confusion. [**H**. *Acyclicity in the definition of MPS*]: In the definition of MPS, $\mathcal{G}_\mathcal{S}$
²⁷ being acyclic is the condition we desire (i.e., a desideratum). In other words, we did not consider a set of actions and
²⁸ their contexts that would create a cycle in its induced graph $\mathcal{G}_\mathcal{S}$. If you can further specify why the definition is unclear,
²⁹ we will be able to address the issue better. Thanks for letting us know recent results about acyclicity-inducing policies
³⁰ on a more general class of graphs. We will look into the connection.
³¹ **Reviewer #2** [**I**. *Practicality of the results*]: As you have mentioned, we made an interesting connection between
³² causality and RL by providing theoretical insights. We expect that, as we better understand such connections, our
³³ theoretical results will guide the design of practical algorithms or applications. [**J**. *Complexity Analysis*]: We consider
³⁴ studying efficient algorithmic characterizations of Thm. 2 and 3 as important research directions. At the moment,
³⁵ these theorems provide graphical understandings of redundancy and optimality. Hence, we do not have algorithmic
³⁶ characterizations for Thm. 2 and 3. [**K**. *Cost of calculation*]: Both the time steps till the convergence and time for
³⁷ calculation seem two important aspects when considering time/computational cost. We will elaborate in the paper the
³⁸ implications of the cost of calculation in evaluating the performance of agents. [**L**. *D-Separation*]: Thm. 2 primarily
³⁹ focuses on whether the variations of some of contexts are ignorable by examining their fixability. In doing so, partly,
⁴⁰ Thm. 2 utilizes the property of d-separation with deterministic relationships. Thm. 2 being a sufficient condition for
⁴¹ non-NRO is orthogonal to d-separation being complete for conditional independence.
⁴² **Reviewer #3** [**M**. *Accessibility to ML audience*]: We tried to utilize figures and examples in most cases to be accessible.
⁴³ We will be able to make the paper more accessible given one additional page for the accepted paper. [**N**. *Causal graph*
⁴⁴ *assumption*]: It is assumed but also can be (partially) inferred from existing data or through interactions. Studying
⁴⁵ under the availability of a causal graph is a necessary step towards better understanding what agents can/should do with
⁴⁶ policies when the agent can only access to partial information about the underlying environment. [**O**. *Reorganizing*
⁴⁷ *preliminaries/Appendix A*]: We are planning to incorporate the essence of Appendix A into introduction so as to better
⁴⁸ motivate the paper. [**P**. *Term 'listening'*]: The term 'listening' is used to carry the meaning of the agent being 'actively
⁴⁹ engaging in observing' variables to determine actions (i.e., to use as a context). Please read also [**G**] in Reviewer 1.
⁵⁰ **Reviewer #4** [**Q**. *Practicality of the results*]: Please check out [**I**] in Reviewer 2. [**R**. *'Surprising results' in the abstract*]:
⁵¹ We intended to emphasize that a policy trying to intervene more (or all) variables blindly, even with utilizing contexts,
⁵² can be failed to converge incurring regret all the times. People often believe that intervening more variables always leads
⁵³ to a better outcome, which is not always the case. (We also agree that 7e is another surprising result since non-ancestors
⁵⁴ of $Y$ in $\mathcal{G}$ are considered irrelevant.) [**S**. *Term 'optimality' in the introduction*]: Thanks for catching our mistranslation
⁵⁵ of an expression $\neg\forall_{\mathcal{M}\sim\mathcal{G}}\exists_{\mathcal{S}'\neq\mathcal{S}}\mu_\mathcal{S}^* \leq \mu_{\mathcal{S}'}^*$, which is the double-negation of Def. 5 (roughly) $\exists_{\mathcal{M}\sim\mathcal{G}}\forall_{\mathcal{S}'\neq\mathcal{S}}\mu_\mathcal{S}^* > \mu_{\mathcal{S}'}^*$.
⁵⁶ We are planning to revise the introduction example with more formal notation (redundancy and optimality) to avoid any
⁵⁷ confusion. A relevant response is also in [**O**]. [**T**. *Thm. 2 is involved*]: We will improve the presentation of Thm. 2 by
⁵⁸ incorporating additional figures such as Fig. 15 in the Appendix. Thank you for the suggestion.

[Meta-Review · NeurIPS 2020]

This paper consider a structural bandit problem where we wish to define a subset of variables to observe and intervene on, such that optimal performance can still be achieved. While the reviewers had some difficulty with some terms in the paper (such as 'simplicity,') this was to a large extent clarified in the rebuttal. The reviewers found this paper to be a nice addition to the growing 'structural bandits' literature.